# Cost-Effectiveness of a Problem-Solving Intervention Aimed to Prevent Sickness Absence among Employees with Common Mental Disorders or Occupational Stress

**DOI:** 10.3390/ijerph17145234

**Published:** 2020-07-20

**Authors:** Marijke Keus Van De Poll, Gunnar Bergström, Irene Jensen, Lotta Nybergh, Lydia Kwak, Caroline Lornudd, Malin Lohela-Karlsson

**Affiliations:** 1Division of Intervention and Implementation Research in Worker Health, Institute of Environmental Medicine, Karolinska Institutet, SE-171 77 Stockholm, Sweden; gunnar.bergstrom@ki.se (G.B.); irene.jensen@ki.se (I.J.); lotta.nybergh@ki.se (L.N.); lydia.kwak@ki.se (L.K.); 2Centre for Musculoskeletal Research, Department of Occupational Health Sciences and Psychology, University of Gävle, SE-801 76 Gävle, Sweden; 3Department of Learning, Informatics, Management and Ethics (LIME), Karolinska Institutet, SE-171 77 Stockholm, Sweden; caroline.lornudd@ki.se; 4Department of Medical Science, Uppsala University, SE-751 85 Uppsala, Sweden; malin.lohela.karlsson@regionvastmanland.se; 5Centre for Clinical Research, Region Västmanland—Uppsala University, Hospital of Västmanland, SE 721 89 Västerås, Sweden

**Keywords:** cost-benefit, cost-effectiveness, problem solving intervention, sickness absence, production loss at work, common mental disorders, stress-related problems, return to work

## Abstract

The cost-benefit and cost-effectiveness of a work-directed intervention implemented by the occupational health service (OHS) for employees with common mental disorders (CMD) or stress related problems at work were investigated. The economic evaluation was conducted in a two-armed clustered RCT. Employees received either a problem-solving based intervention (PSI; *n* = 41) or care as usual (CAU; *n* = 59). Both were work-directed interventions. Data regarding sickness absence and production loss at work was gathered during a one-year follow-up. Bootstrap techniques were used to conduct a Cost-Benefit Analysis (CBA) and a Cost-Effectiveness Analysis (CEA) from both an employer and societal perspective. Intervention costs were lower for PSI than CAU. Costs for long-term sickness absence were higher for CAU, whereas costs for short-term sickness absence and production loss at work were higher for PSI. Mainly due to these costs, PSI was not cost-effective from the employer’s perspective. However, PSI was cost-beneficial from a societal perspective. CEA showed that a one-day reduction of long-term sickness absence costed on average €101 for PSI, a cost that primarily was borne by the employer. PSI reduced the socio-economic burden compared to CAU and could be recommended to policy makers. However, reduced long-term sickness absence, i.e., increased work attendance, was accompanied by employees perceiving higher levels of production loss at work and thus increased the cost for employers. This partly explains why an effective intervention was not cost-effective from the employer’s perspective. Hence, additional adjustments and/or support at the workplace might be needed for reducing the loss of production at work.

## 1. Introduction

Common mental disorders (CMD, i.e., depression, anxiety and adjustment disorders) are highly prevalent and are associated with a total cost estimated to be more than 4% of the GDP across the EU countries. Of this cost, only 1.3% of the GDP is spent directly on health care, while the rest is comprised of indirect costs associated with social security programs, lower employment on the labor market and production loss among employees that are sick at work or on sickness absence due to CMD [1].

Interventions that include the workplace have been shown to be effective for preventing, or reducing, sickness absence among employees with CMD [2]. Work-directed interventions try to adjust the work environment (e.g., change of responsibilities/assignments; change in work schedule) to facilitate return to work (RTW), or to maintain work ability. Further, it tries to help the employee to manage his/her psychiatric symptoms to reduce or prevent sickness absence [2]. Hence, work-directed interventions should lead to better health, improved work ability and reduced costs related to sickness absence and production loss at work, i.e., reduced production due to ill health while at work.

Research has shown that work-directed interventions given at the OHS which are based on problem solving therapy or cognitive behavioral therapy for employees on sickness absence for CMD can decrease time to first return to work (i.e., partial RTW). However, these interventions did not decrease days until full RTW [3,4,5,6]. Further, among studies that investigated outcomes of RTW (e.g., lost time, work functioning and costs related to work disability), cognitive behavioral therapy programs with a focus on work-relevant solutions had a moderate to strong level of evidence for being effective for employees with mental health conditions [7].

In Keus van de Poll et al. [8], we compared a work-directed intervention based on problem-solving (PSI) with care as usual (CAU), both conducted by the occupational health service (OHS) for employees with CMD to prevent, or reduce, sickness absence and facilitate RTW. Long-term sickness absence was reduced with at least 15 days over a one-year follow-up, for employees that received PSI. Further, employees that received PSI had an earlier partial RTW compared to employees that received CAU, but there were no differences for full RTW. Conclusively, PSI appeared to be effective in decreasing long-term sickness absence and facilitating RTW when compared to CAU. These results expand the evidence that work-directed interventions with a problem-solving strategy are more effective compared with CAU in decreasing sickness absence or promoting RTW among employees with CMDs or stress-related symptoms at work [9,10].

As resources for occupational health interventions are scarce [11], not only the effect but also the cost-effectiveness of the interventions might constitute an important incentive for employers and public policymakers to adopt them [12]. From the employer’s perspective, an intervention might result in less benefits than costs, i.e., not being cost-effective, whereas from the societal perspective, it indicates to be cost-effective when including the costs and benefits across all stakeholders (e.g., employers and societal institutions). This could be because from the employer’s perspective, only costs and consequences borne by the employer’s should be included, whereas all costs and consequences are considered from the societal perspective, irrespective of who pays or benefits from it [12]. Therefore, it is recommended that studies include various types of economic evaluations in conjunction with analysis on effectiveness, in order to inform all relevant stakeholders [13].

Previous studies that have evaluated effectiveness and cost-effectiveness on interventions aimed to promote RTW and/or prevent or reduce mental health problems for employees with CMD, showed that there is no clear evidence for the (cost-) effectiveness [14]. However, as work-directed interventions have shown to be effective in reducing sickness absence and promoting RTW [7], it might be that such interventions not only are more effective but also more cost-effective than CAU. Since the work-directed intervention studied by Keus van de Poll et al. [8] showed to be effective in reducing sickness absence and promoting RTW, the aim of the current study was to investigate the cost-benefit and cost-effectiveness of this work-directed intervention from both an employer and a societal perspective.

## 2. Materials and Methods

### 2.1. Study Design and Setting

An economic evaluation of an intervention given by three different OHS (one nationwide with two units and two regional OHS) to employees with or at risk for CMDs was conducted from an employer and societal perspective within a two-armed clustered RCT with a one-year follow-up. Employees in the experimental group received PSI, which was compared to CAU given at the OHSs. Both interventions were work-directed though with different structure and content.

The Swedish OHS operates on the open market, independently from the state funded health care system. An employer must ensure the availability of relevant OHSs. The OHS works for the prevention and elimination of health risks at the workplace. OHSs has knowledge of the employee’s work environment and can offer interventions to prevent sickness absence that take into account both the individual and the workplace [15].

Figure 1 shows the recruitment of consultants, intervention and control groups (Figure 1 is copied without changes from Keus van de Poll et al. [8] under a CC BY 4.0 license). Randomization took place at the OHS consultant level using computer-generated random numbers randomizing OHS consultants into giving either PSI or CAU. Randomization was stratified by OHS unit. Employees were recruited by consultants at the participating OHSs between August 2015 and June 2017. Inclusion criteria were that the employee sought help at the OHS for a new episode of occupational stress or symptoms related to CMDs affecting the ability to work. If the employee was on sickness absence due to CMDs this period should not have exceeded three months. The employee should agree to the involvement of the employee’s manager in the intervention, and the employee had to understand both written and spoken Swedish. Employees, but not the consultants, were blinded to the possibility of receiving another intervention within the trial. This was because the OHS consultants were pre-randomized into delivering one of the two interventions.

Data about the employees at baseline and about the professions and gender of the OHS consultants is presented in Table 1; Table 2. Ethical approval was obtained from the Ethical Review Board in Stockholm (registration number 2015/549-31/1). More details of the study design and procedures have been reported elsewhere [8,16].

### 2.2. Problem-Solving Intervention (PSI)

Members of the research group and a clinical psychologist offered the OHS consultants a 1-day training course in the intervention. Consultants also received detailed work sheets. The intervention was primarily focused on adjusting the work situation and secondarily on advising the employee about stress management. The goal of the intervention was to promote work ability and RTW. The frameworks of the intervention are based on Problem-Solving therapy [17] and the mismatch model concerning the match between the employee and the work environment [18]. In line with the mismatch model, six aspects of the work situations were addressed during the meetings in the intervention (i.e., workload, control, reward, community, fairness, values). The intervention followed a manual consisting of three steps and a follow-up; an interview with the employee’s manager and the employee, respectively (steps 1 and 2), and a joint meeting between the consultant, the employee’s manager and the employee to actively work with problem solving concerning the work situation (step 3) [9,10]. At least three follow-ups of the manager and the employee during a 3-month period were recommended.

### 2.3. Care as Usual

Consultants that were randomised to CAU received a general introduction in research about psychosocial factors and mental health at work for approximately 1 h. CAU at the participating OHS was a work-directed intervention that implied involvement of both employee and manager in the process but not with the same structure and/or content as in PSI. Neither was CAU deliberately based on the same frameworks (problem-solving therapy and the mismatch model) as PSI [16].

### 2.4. Data Collection

Data regarding registered sickness absence, defined as the total number of net absence days from work due to sickness during the 12-month follow-up, was provided by the Swedish Social Insurance Agency (SSIA). For each period of sickness absence exceeding 14 days, dates, degree and number of days of sickness absence were registered. Data regarding short-term sickness absence (up to 14 days) is not registered by the SSIA. To be able to collect data regarding short-term sickness absence and production loss due to ill health [19,20], questions were sent as SMS messages to the participants every fourth week during the 12-month follow-up. The follow-up period started immediately after baseline, and the first follow-up measurements were sent out four weeks after baseline, covering these four weeks. Response rates on the SMS question ranged between 81% and 98% per measurement for CAU and between 85% and 100% per measurement for PSI. Further, questionnaires including questions regarding ordinary working hours and other interventions that employees received beside PSI or CAU during the follow-up were administered at three occasions (i.e., at baseline, six months and 12 months after study start). See Appendix A for an overview of the questions and information regarding other interventions that employees received beside PSI or CAU.

### 2.5. Economic Evaluation

A cost benefit analysis (CBA) and a cost-effectiveness analysis (CEA) were performed from both an employer and societal perspective. The CBA was used to compare intervention costs with monetary benefits (e.g., potential savings) of reduced production loss (i.e., sickness absence and production loss at work), whereas the CEA was used to compare intervention costs, including costs for production loss at work and short term sickness absence, with the total number of net sickness absence days.

### 2.6. Intervention Cost

The cost of the intervention was calculated based on an average fee per hour paid to the OHS for each meeting plus costs related to travel time to/from the meetings and time to attend the meetings. Travel time was standardized to 1 h per meeting (30 min/single way) [16], whereas time in meetings was based on treatment time (Table 3). Cost for lost time due to treatment or travel was calculated using national median wages (33,700 SEK for year 2017/3390 Euros) [21,22], including general payroll taxes (31.42%) [23], to reflect the minimum cost for the employer when the worker did not participate in their regular production activities.

### 2.7. Production Loss

Potential benefits of the intervention were calculated using the number of sickness absence days as well as production loss due to ill health. Total sickness absence costs were calculated using both short- and long-term periods of sickness absence according to the Human Capital Approach [11]. In Sweden, employers pay 80% of the wages for sickness absence during days 1–14 and 10% for days 15–90. The SSIA covers the cost for sickness absence from day 15 and days beyond this period [24]. Hence, for analyses from the employer’s perspective, 80% of the daily wage was calculated for short-term sickness absence (days 1–14), and 10% was calculated for long-term sickness absence, i.e., between days 15–90. The total number of short-term sickness absence days was calculated in two steps. First, for each individual, the first 14 days of each new long-term sickness period in the SSIA data that started within the one-year follow-up was counted as short-term sickness days. Next, we compared self-registered sickness absence data with the data from the SSIA. Self-registered periods <14 days that did not overlap with sickness periods registered by the SSIA were counted as short-term sickness absence days. Long-term sickness absence was calculated by counting the total number of net days within the 15–90 days interval. We added all days up to 90 net days for each uninterrupted period of sickness absence registered in the SSIA database within the one-year follow-up for all employees within PSI and CAU.

For the analyses from the societal perspective, costs were based on 100% of the wages for all sickness absence days (short-term and long-term) during the follow-up period. Number of days for short-term sickness absence was similar to days from the employer’s perspective. The total number of long-term sickness days (>14 days) was calculated by adding the number of all net days that fell within the one-year follow-up for all employees within PSI and CAU, with help of the register data from the SSIA.

Production loss due to ill health was used to calculate the costs for reduced production at work. To identify the actual hours at work, each month the following formula was used:*Working hours per month* − *hours lost due to sickness absence* = *Number of hours at work per month*

The remaining hours were multiplied with production loss to identify the number of hours lost. This formula was used to avoid double counting of costs due to sickness absence and reduced performance while at work. Thereafter, the cost of production loss was calculated using the numbers of hours of lost production multiplied with hourly wages. National median wages and payroll taxes were used to estimate the costs of production loss.

All costs and consequences were converted to a single year (2015) using the consumer price index [25]. Discount of costs or consequences has not been conducted since it is not needed for a short follow-up period of one year [11].

### 2.8. Data Analysis

For all analyses, we used bootstrapping method with 5000 replications to calculate differences in cost (ΔC) and differences in effect measure (ΔE).

#### 2.8.1. Employer Perspective

All costs related to the intervention paid by the employer were included in the calculation. Costs due to travel and meeting time were included only for workers not on sickness absence. The mean cost in each treatment group was used to calculate the mean incremental cost, i.e., the differences in the mean cost between PSI and CAU.

The potential benefit of reduced production loss was calculated using the sickness absence costs paid for by the employer and the cost of reduced production due to ill health. The cost-effectiveness was evaluated using CBA and CEA.

The effect measure in the CEA was total number of registered sickness absence days. The result was presented as incremental cost-effectiveness ratio (ICER).

#### 2.8.2. Societal Perspective

The economic evaluation from the societal perspective included all costs and benefits relevant to the society. Besides costs related to the intervention, potential benefits of reduced production loss were calculated. This included both short-term and long-term sickness absence costs and costs due to production loss while at work.

The cost-effectiveness was evaluated using CBA and CEA. The CBA was used to calculate the net benefit, whereas the CEA was used to calculate the cost for an additional sickness absence day, presented as ICER.

#### 2.8.3. Sensitivity Analysis

To evaluate the sensitivity of the result of the cost-effectiveness analyses, two sensitivity analyses were performed. These excluded costs due to production loss while at work (sensitivity analysis 1) and costs due to production loss while at work together with short-term sickness absence (sensitivity analysis 2). The reason for this was that employees in PSI returned to work sooner compared to employees in CAU [8]. This means that employees in PSI are “at work” for more days during the follow-up than employees in CAU and potentially have more hours of reduced production and more days of short-term sickness absence. These costs affect the employers but are not present when the employees are on long-term sickness absence. Furthermore, they could affect the total costs for production loss at work as well as cost for short-term sickness absence. Another reason was that both are subjective measures, which implies a risk for recall bias.

## 3. Results

### 3.1. Costs for PSI and CAU

Table 4 shows the mean costs for PSI and CAU from an employer and societal perspective. Total costs for intervention and travel time were higher for CAU compared to PSI, from both the employer and societal perspective.

### 3.2. Production Loss

Table 4 shows that costs for short-term sickness absence were higher for PSI compared to CAU, whereas costs for long-term sickness absence were lower for PSI compared to CAU, from both the employer and societal perspective.

### 3.3. Cost-Benefit and Cost-Effectiveness Analyses—Employer Perspective

According to the CBA in Table 5, PSI is not cost-effective compared to CAU. The negative net benefit for PSI is mainly due to higher short-term absence costs and cost related to production loss due to ill health. The ICER for CEA in Table 5 shows that the cost for each day of reduced long-term sickness absence day is higher for PSI compared to CAU. The difference in cost is mainly due to higher short-term absence costs and related to production loss due to ill health. This indicates that PSI is not cost-effective from an employer perspective.

### 3.4. Sensitivity Analyses

Table 6 shows the results for the sensitivity analyses. Exclusion of the costs for production loss (sensitivity analysis 1) did not change the interpretation of the result, neither for the employer nor for society. When both costs for short-term sickness absence and production loss at work were excluded (sensitivity analysis 2), PSI was cost-beneficial from the employer’s perspective. The result from the societal perspective did not change, i.e., PSI was still found cost-effective. Further, the ICER for CEA changed, i.e., PSI was both more effective and less costly than CAU. This was the case from both the employer and societal perspective.

## 4. Discussion

The aim of the current study was to investigate the cost-effectiveness of work-directed interventions conducted by the OHS. Both the employer and societal perspective were considered. The result indicated that PSI was a cheaper intervention than CAU but that costs related to short-term sickness absence and production loss due to ill health, i.e., costs that occur while employees are at work, were higher for PSI, whereas the cost related to long-term sickness absence was higher for CAU. The interpretation of cost-effectiveness varied for the different perspectives and depended on the analysis conducted. PSI was not considered cost-effective from the employer’s perspective. The sensitivity analysis indicated that this was due to costs related to higher short-term sickness absence and health-related production loss. The CBA from the societal perspective indicated that PSI was cost-beneficial. A one-day reduction in long-term sickness absence costed on average € 101, a cost that primarily was borne by the employer.

From a societal perspective, having healthy and productive employees with low sickness absence is desirable. Hence, PSI can contribute to reaching these goals, since the results in Keus van de Poll et al. [8], showed that PSI reduced days of long-term sickness absence and that employees that received PSI returned to work sooner compared to CAU, measured over a one-year follow-up. Reasonably, PSI should decrease costs for long-term sickness absence. Indeed, this was found for PSI, resulting in a decreased socio-economic burden compared to CAU. This is in contrast to previous cost-effectiveness studies evaluating RTW interventions for employees with CMD, where no evidence for cost-effectiveness from the societal perspective was found [14]. Part of the explanation was that the interventions had no effect on RTW rates or production loss costs. However, the current study indicated that implementation of PSI, a work-directed intervention for employees with CMD, should be recommended to the policy makers.

When employees return to work sooner, as found in Keus van de Poll, et al. [8], sickness absence costs decrease, which is beneficial for society. However, the economic evaluation from the employer’s perspective indicated higher sickness absence costs for PSI compared to CAU. Similar results were found in a previous systematic review [14], i.e., RTW-interventions were not cost-effective from the employer’s perspective. The difference with the current study is that the results in Hamberg-van Reenen et al. [14] were mainly explained by ineffective interventions, while a potential explanation in the current study is that employees at work had more short-term sickness absence days and reduced production at work when they received PSI compared to CAU. Costs for short-term sickness absence and production loss at work are borne by the employer but not when the employees are absent due to long-term sickness. Since PSI was more effective than CAU, short-term sickness absence costs and costs related to lost production will be added as a cost for employers to a larger extent for PSI than for CAU. This can explain why PSI was not cost-effective from the employers’ perspective. Additional calculations were performed for production loss at work in an attempt to understand the reason behind the higher costs in PSI. Only the average loss for employees who were present at work was calculated. This was done using total hours of production loss reported per year in relation to total working hours per year for each group, i.e., sickness absence hours were excluded. The result indicated that the average production loss for those present at work was similar for both groups (PSI 30,0% production loss, CAU 31,5% production loss). The reason for a higher average cost of production loss in PSI seems to be because employees are present at work for more hours during the follow-up year in PSI (total working hours) compared to CAU, resulting in a higher total cost for production loss in the group, which in turn results in higher average costs for that group. When conducting economic evaluations of cost-effectiveness studies of RTW studies, this problem needs to be dealt, which might require development of methods used to conduct such calculations.

Further, costs for short-term sickness absence and production loss at work are not unique for employees with CMD or stress related problems. Reasonably, employees without any symptoms of (mental) ill health also have short-term sickness absence days and production loss at work resulting in costs for employers. Hence, there might be an average cost for all employees at work. Employees that return to work often replace a substitute with an average short-term sickness absence and reduced production. Therefore, a more correct way to calculate costs, or savings, related to RTW might be to only include the actual difference in cost related to production loss between an average employee at work and the employee who returns to work. To be able to do that, information about average production loss and short-term sickness absence in a working population is required. Future studies should consider this phenomenon when conducting an economic evaluation of a work-directed intervention from the employer’s perspective.

Another explanation for the higher costs related to production loss might be that adaptations in the work environment could have helped employees returning to work earlier, but remaining symptoms of mental ill health might have caused production loss at work. That employees have reduced production due to ill health directly after RTW has been shown in other studies [26,27]. For example, in a study on employees previously on sickness absence due to musculoskeletal disorders, lower production was still found 12 months after the sickness absence ended [27]. This suggests that additional adjustments and/or support might be needed at the workplace for employees after RTW to reduce production loss and related costs.

Approximately half of the included participants in the current study were at risk for sickness absence due to CMD when receiving the intervention. For them, both PSI and CAU was used as a preventive intervention. Previous studies have found that worksite interventions designed to prevent or treat mental health problems might be cost-effective from both the employer’s and societal perspectives [14]. Since the study population in the current study was relatively low, it was not possible to conduct subgroup analyses separating employees at risk of sickness absence from those currently on sickness absence. Sickness absence is often preceded by sickness presenteeism, resulting in production loss to the company. Preventive interventions are often designed to prevent sickness absence. However, it is possible that they reduce the level of production loss for those participating in the intervention and increase production. Future studies are suggested to evaluate the effect of production loss as well as the cost-effectiveness of PSI separately as a preventive and rehabilitating intervention.

Sickness absence due to CMDs is a major challenge connected to high costs in many countries [1]. Hence, the results should be of interest from an international perspective. Since our results in general are in line with results from other studies [5], there appears to be some generalizability across countries. However, since the social insurance system differs among countries, the cost-effectiveness of a similar intervention as PSI might vary. To illustrate this, employers in Sweden only have to pay 80% of the wages for days 1–14 and 10% for days 15–90 of long-term sickness absence [24]. In the Netherlands, employers pay at least 70% of the wages during the first year and 70% during the second year [28]. Hence, costs of long-term sickness absence for an employer in the Netherlands will be higher compared to Sweden, due to the different social insurance system.

### Methodological Considerations

A strength of the current study is that despite the risk for recall bias in the subjective data on self-registered sickness absence, we were able to make a distinction between short-term and long-term sickness absence, which allowed us more realistic calculations of costs for employer and society. In Sweden, this distinction between short-term and long-term sickness absence is of value due to the national social insurance system, in which the employer pays for short term sickness absence. In other countries with other social insurance systems, the value of such a distinction might differ.

It should be considered that cost calculations for PSI are based on the number of sessions and session time prescribed in the manual consultants in PSI received. However, some of the consultants spent more time on PSI than what was prescribed. In CAU, the structure (e.g., number of sessions and session time) varied across the OHS units and within some OHS units could CAU be individually adapted. The calculations for CAU were based on an estimation of the mean number of sessions and session time for each unit (Table 3). For both PSI and CAU, travel costs were based on standardized travel times. Further, costs for follow-up meetings, other interventions that the employer received beside PSI or CAU and costs for substitute personnel and reduced production for the employer while employees were on sickness absence can affect the result of the cost-effectiveness analyses. Since data regarding these issues was incomplete or not possible to use to calculate additional health care costs, these costs were not included in the economic analyses. Consequently, costs and/or savings for both CAU and PSI might be underestimated. Moreover, national median wages were used to calculate for example sickness absence costs and costs related to production loss due to ill health. As the study population consisted of employees in various occupations, this might under- or overestimate the actual costs and potential savings. Future studies should consider including additional costs and benefits relevant for the employer and society when evaluating the cost-effectiveness of RTW or sickness absence.

A limitation with the current study is the relatively low power. Based on power calculations, the target was to recruit 150 employees for the study, but the final sample consisted of 100 employees. This might also have had impact on the result of the cost-effectiveness.

## 5. Conclusions

A structured work-directed intervention based on problem solving for employees with CMD leads to fewer long-term sickness absence days and a faster RTW. Furthermore, it reduces the socio-economic burden compared to receiving care as usual. The results of this study indicate that the implementation of such an intervention for employees with CMD should be recommended to policy makers. However, employees with CMD perceive a high level of production loss while at work, which results in a cost to the employers. This cost related to production loss due to ill health while at work is a major explanation for why an effective intervention is not considered to be cost-effective from the employer’s perspective. Additional adjustments and/or support at the workplace might be needed for employees after RTW to reduce production loss and related costs.

## Figures and Tables

**Figure 1 ijerph-17-05234-f001:**
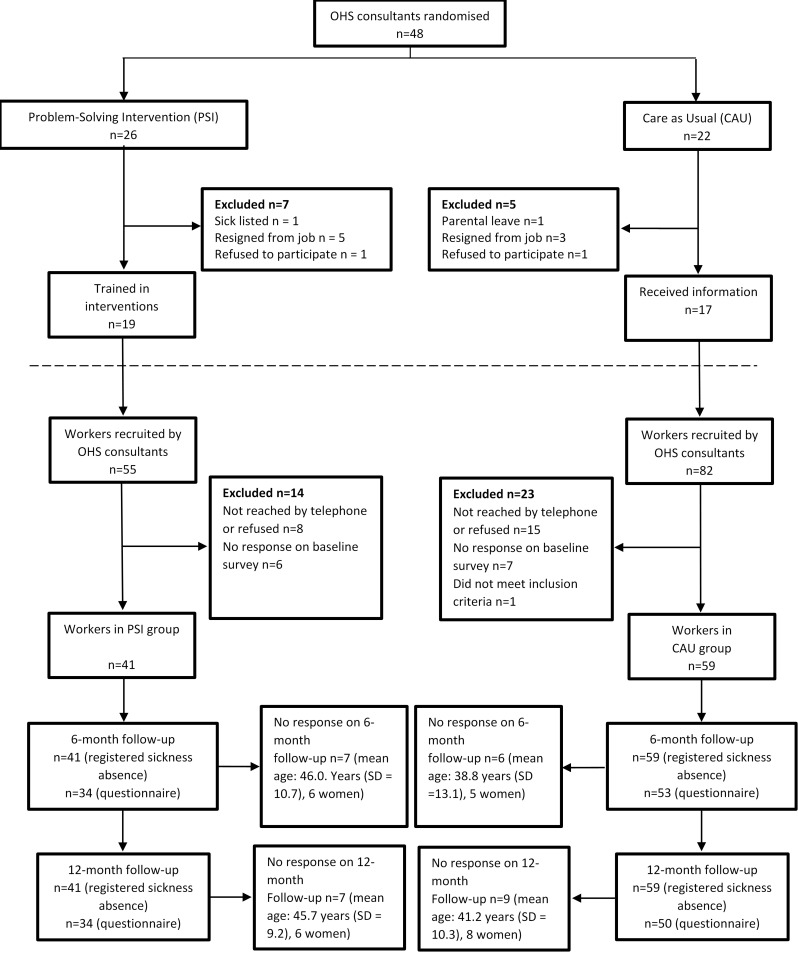
Flow diagram showing the recruitment of consultansts, experimental and control groups.

**Table 1 ijerph-17-05234-t001:** Employee Characteristics per Study Group at Baseline.

Sociodemographic Characteristics		PSI (*n* = 41)		CAU (*n* = 59)	
Age, years, m (sd)		42.7	(10.4)	44.0	(9.6)
Female, n (%)		37	(90)	43	(73)
Children, n (%)		23	(56)	39	(66)
Education level, n (%)					
	Prim./sec. education	14	(34)	20	(34)
	Higher education/university	27	(66)	39	(66)
Ordinary working hours					
	Full time (40 h/week)	31	(76)	52	(88)
	Part time (< 40 h/week)	10	(24)	7	(12)
Employer, n (%)					
	Municipality, county, state ^1^	38	(93)	39	(66)
	Private business	3	(7)	20	(34)
Profession, n (%)					
	Teacher	9	(22)	2	(3)
	manager	2	(5)	8	(14)
	Assistant nurse	4	(10)	5	(8)
	nurse	3	(7)	6	(10)
	IT-architect/system development/test leader	1	(2)	4	(7)
	Administrative (executive) official	2	(5)	2	(3)
	Caretaker/personal assistant	2	(5)	2	(3)
	Client support	0	(0)	4	(7)
	Other ^2^	18	(44)	26	(45)
Registered sickness absence, n (%)					
	No sickness absence	20	(49)	31	(53)
	Sickness absence	21	(51)	28	(47)
% Production loss due to ill health		62	(28)	65	(24)

PSI = problem-solving intervention; CAU = care as usual. ^1^ Three individuals in PSI and seven employees in CAU were employed by the state. ^2^ Other professions (*n* ≤ 3 per profession), e.g., child-care workers, engineers, psychologists, physiotherapists, biomedical analysts, project leaders, developers, consultants, welfare officer.

**Table 2 ijerph-17-05234-t002:** Profession and Gender for the OHS Consultants in *n* (%).

	PSI *n* = 26	CAU *n* = 22
Female	21 (80.8)	19 (86.4)
Male	5 (19.2)	3 (13.6)
Nurses and ergonomists	16 (61.5)	17 (77.3)
Behavioral scientists and psychologists	8 (30.8)	4 (18.2)
Physician	2 (7.7)	1 (4.5)

PSI = problem-solving intervention; CAU = care as usual.

**Table 3 ijerph-17-05234-t003:** CAU at the Different OHS Units.

		Attending the Meeting	Duration in Minutes (Approx.)	Location
PSI				
	Step 1	OHS consultant and manager	30	phone
	Step 2	OHS consultant and employee	90	OHS
	Step 3	OHS consultant, employee and employer	60	OHS
CAU				
	Units 1 and 2			
Step 1		OHS consultant and manager	30	Phone
Step 2		OHS consultant and employee	120 *	OHS
Step 3		OHS consultant, employee and manager	60 *	OHS
	Unit 3			
Step 1		OHS consultant, employee and manager	60	OHS
Step 2		OHS consultant and employee	120–210 **	OHS
	Unit 4			
Step 1		OHS consultant and manager	30	phone
Step 2		OHS consultant, manager and employee	90	OHS

PSI = problem-solving intervention; CAU = care as usual; OHS = occupational health service. * (incl. 15 min administration/reporting time. ** 2–3 meetings; in total 2–3.5 h.

**Table 4 ijerph-17-05234-t004:** Total and Mean Component Costs (EURO/Employee) for PSI and CAU from an Employer and a Societal Perspective.

	PSI	95% CI		CAU	95% CI	
Component		Lower; Upper			Lower; Upper	
*Employer’s perspective*						
Intervention Costs						
Intervention	405	393; 418		445	426; 463	
Travel time	62	52; 72		64	54; 75	
Total	467	445; 489		509	481; 537	
Sickness absence Costs						
Long-term sickness absence ^1^	1358	878; 1838		1647	1218; 2077	
Short-term sickness absence ^2^	1876	1246; 2506		1310	797; 1823	
Total	3234	2498; 3970		2957	2306; 3609	
Presenteeism costs						
Production loss due to ill health	16,101	13,577; 18,625		13,942	11,576; 16,308	
Total cost for production loss ^3^	19,335	16,920; 21,750		16,900	14,391; 19,408	
*Societal perspective*						
Intervention Costs						
Intervention	446	446; 447		494	477; 511	
Travel time	95	95; 95		101	94; 107	
Total	541	541; 542		594	572; 617	
Sickness absence costs						
Long-term sickness absence ^4^	16,409	9372; 23,445		24,622	17,160; 32,084	
Short-term sickness absence ^2^	2280	1514; 3045		1592	968; 2216	
Total	18,688	11,712; 25,665		26,214	18,874; 33,554	
Presenteeism costs						
Production loss due to ill health	16,101	13,577; 18,625		13,942	11,576; 16,308	
Total cost for production loss ^3^	34,789	27,551; 42,027		40,156	33,205; 47,107	

PSI = problem-solving intervention; CAU = care as usual. ^1^ Mean (sd) number of sickness absence for the period between day 15–90 (net days): 28.1 (30.0) for PSI and 32.7 (31.8) for CAU. ^2^ Mean (sd) total number of short-term sickness during the one-year follow-up (days): 9.0 (9.6) for PSI and 6.3 (9.2) for CAU. ^3^ Total cost for production loss (total cost for sickness absence + cost for production loss due to ill health). ^4^ Mean (sd) total number of long-term sickness during the one-year follow-up (net days): 64.5 (90.3) for PSI and 92.8 (116.1) for CAU.

**Table 5 ijerph-17-05234-t005:** Incremental Cost, Effect and ICERs for the Different Analyses, for Both Employer and Societal Perspective. All Numbers are Based on Bootstrap with 5000 Repetitions.

		C *		E *					
		CAU	PSI	CAU	PSI	ΔC	ΔE	ICER	NMB
CBA	*employer*	508	467	16,892	19,342	41	−2450		−2409
	*society*	594	541	40,178	34,701	53	5477		5530
CEA	*employer*	15,780	18,430	93	65	−2650	28	−95	
	*society*	16,109	18,930	93	65	−2821	28	−101	

CAU = care as usual; PSI = problem-solving intervention; CBA = cost-benefit analysis; CEA = cost-effectiveness analysis; C = incremental cost in Euro; E = incremental effect in Euro (CBA) or days (CEA); ΔC = difference in incremental cost for CAU and PSI; ΔE = difference in incremental effect for CAU and PSI; ICER = incremental cost-effectiveness ratio; NMB = net monetary benefit. * For all incremental costs and incremental effects, differences between PSI and CAU were statistically significant, *p* < 0.001.

**Table 6 ijerph-17-05234-t006:** Incremental Cost, Effect and ICERs for the Sensitivity Analysis, from both an Employer and a Societal Perspective.

		C		E		ΔC	ΔE	ICER	NMB
		CAU	PSI	CAU	PSI				
Sens.analysis 1 ^1^									
CBA	*employer*	509	467	2957	3225	42	−268		−226
	*society*	595	541	26,306	18,712	54	7594		7648
CEA	*employer*	1820	2348	93	65	−528	28	−19	
	*society*	2189	2824	93	65	−635	28	−23	
Sens.analysis 2 ^2^									
CBA	*employer*	509	467	1647	1358	42	289		331
	*society*	594	541	24,605	16,358	53	8247		8300
CEA	*employer*	509	467	93	65	42	28	1.5	
	*society*	594	541	93	65	53	28	1.9	

CAU = care as usual; PSI = problem-solving intervention; CBA = cost-benefit analysis; CEA = cost-effectiveness analysis; C = incremental cost in Euro; E = incremental effect in Euro (CBA) or days (CEA); ΔC = difference in incremental cost for CAU and PSI; ΔE = difference in incremental effect for CAU and PSI; ICER = incremental cost-effectiveness ratio; NMB = net monetary benefit. ^1^ Costs for production loss were excluded. ^2^ Costs for production loss and short-term sickness absence were excluded.

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
