# Peer review of "Cost-Effectiveness of a Problem-Solving Intervention Aimed to Prevent Sickness Absence among Employees with Common Mental Disorders or Occupational Stress"

_ijerph, 2020, doi:10.3390/ijerph17145234_

Round 1

Reviewer 1 Report

Thank you for the opportunity to review this interesting ms. Overall the topic is really good, fresh and needed, but the ms needs more clarifications and limitation discussion. Here are my current comments.

Headline:

  • Currently it is really long and not entirely in line with the style of the ms, can it be revised and shortened?

Abstract:

  • Structural abstract would be more clear to read.
  • n should be included in the abstract.
  • “Intervention costs were lower for PSI than CAU”, what does this mean, as CAU had no intervention?
  • Revise “Costs for long-term sickness absence costs”
  • “However, reducing long-term sickness absence, i.e. increased return to work”, could be revised.

Introduction:

  • Employer perspective and societal perspective could be introduced more clearly.
  • Meaning of production loss could be explained.
  • Lines 54-55, “Research on work-directed interventions for employees having CMD has found that such interventions reduce partial but not full RTW”, open these results a little bit more. Also, reduce RTW, is it correct?
  • Lines 55-58, reference 7 has reviewed studies from 1990 to 2015, is there more recent evidence in addition that should be cited? Reference 7 is not all about mental health, clarification could be added.
  • Line 70, “when including the costs and benefits across all stakeholders”, clarify the stakeholders.

Material and methods:

  • Swedish OHS system could be explained shortly, as the OHS systems are very different across countries.
  • Line 85, “with or at risk for CMDs”, how about the stress related problems? In addition, how were these measured (risk or prevalence)?
  • Lines 96-97, “Employees, but not the consultants, were blinded to the possibility of receiving another intervention within the trial.” could be more clear.
  • Who were the employees? How many? What is their socioeconomic background? What kind of employers? What is the professional background of OHS consultants? These should be presented early in the material and methods.
  • Data collection: data loss should be mentioned.
  • Lines 124-126: “The economic evaluation was conducted from both the employer and the societal perspective. A cost benefit analysis (CBA) and a cost effectiveness analysis (CEA) were performed from both an employer and societal perspective.” Repeating could be revised.
  • Line 126, Monetary benefits, be more specific?
  • Doesn´t the employers have records of the short-term sickness absence days? Those would be more reliable.
  • What is production loss? How it´s different from productivity loss? Or from self-rated productivity loss? These should be more clear. In addition, stressed or employees having CMD symptoms may not be as productive at work as they would normally be, so does the calculation hold this perspective or is it just work hours?
  • Lines 134-137 “Cost for lost time due to treatment or travel was calculated using national median wages (33 700 SEK for year 2017/3 390 Euros), including general payroll taxes (31.42%) [22], to reflect the minimum cost for the employer when the worker did not participate in their regular production activities.” This is problematic, it´s too general considering the sample is small and socioeconomically focusing on higher end. Why this was not asked from the employees? Or employer records used?
  • Based on Table 1 the intervention and CAU are not that different, this should be explained more. In addition, the intervention / CAU units are not presented clearly enough in the text. In addition, the content of the CAU should be more closely explained. Also the intervention should be explained in more detailed.
  • The travel time is very short, are these in-house OHS units?
  • From what point did the follow-up start?
  • What about the other visits in OHS and their costs?

Other comments:

  • Participants should be presented in the material and methods, not results.
  • All the limitations noted above should be more closely discussed in the methodological considerations.
  • Sickness absence, absenteeism, sick-listed and sick leave are all used, for conceptual clarity only sickness absence should be used. Same applies to implemented/conducted (intervention).
  • Lines 333-335, these limits are Sweden-specific, should be noted.

Reviewer 2 Report

The paper is an important study about how to reduce work-place sickness rates. It is very detailed and thorough. 

The limitations of the study are acknowledged, including the small size, but I think the authors could be clearer that the intervention did show an important effect on the long term sickness rates, and arguably this is the most significant part of their findings- long term sickness rates are the key factor in predicting loss from the workplace, unemployment and ill health- and the initial literature review could make more of this important point. 

The CEA looking at long term sickness absence is arguably the most important analysis, as to some extent the costs that fall on society and employers from short term sickness will depend on the local employment and social security arrangements in different states or countries. The inclusion of self-reported productivity losses is debated- and given the potential self reporting biases, alongside the very varied productivity of workers is I think quite limited.

I think the research is valid and important, but the key messages need to a little clearer.

Reviewer 3 Report

Dear Authors,

Your article on Preventing sickness absenteeism among employees with common mental disorders or stress-related symptoms at work is a interesting work and I have some comments that authors should address before publication in the Journal.

Lines 49-50 "Work-directed interventions try to adjust the work environment" how? I suggest to explain better

Line 359 "which results in a cost to the employers. This cost " I suggest to explain better what does this cost contain

Round 2

Reviewer 1 Report

Thank you for revising the manuscript and being very thorough in answering the numerous comments made. The clarifications were needed, the manuscript has improved and is now much more clear to the international audience. I warmly recommend publication and wish the authors a good and sunny summer.